# Isolation and Characterization of Bacteriophage VA5 against *Vibrio alginolyticus*

**DOI:** 10.3390/microorganisms11122822

**Published:** 2023-11-21

**Authors:** Qingfang Hao, Yue Bai, Haolong Zhou, Xiuli Bao, Huanyu Wang, Lei Zhang, Mingsheng Lyu, Shujun Wang

**Affiliations:** 1Jiangsu Key Laboratory of Marine Bioresources and Environment/Jiangsu Key Laboratory of Marine Biotechnology, Jiangsu Ocean University, Lianyungang 222005, China; qfhao@jou.edu.cn (Q.H.); yuebai@jou.edu.cn (Y.B.); xlbao@jou.edu.cn (X.B.); wanghuanyuhk@163.com (H.W.); mslyu@jou.edu.cn (M.L.); 2Co-Innovation Center of Jiangsu Marine Bio-Industry Technology, Jiangsu Ocean University, Lianyungang 222005, China; 3Key Laboratory of Special Pathogens and Biosafety, Wuhan Institute of Virology, Chinese Academy of Sciences, Wuhan 430207, China; zhouhaolong221@mails.ucas.ac.cn

**Keywords:** whole-genome sequencing, double-agar plate method, biological characteristics, biological control

## Abstract

Bacteriophages, or phages, can be used as natural biological control agents to eliminate pathogenic bacteria during aquatic product cultivation. Samples were collected from seafood aquaculture water and aquaculture environmental sewage, and phage VA5 was isolated using the double-layer agar plate method, with *Vibrio alginolyticus* as the host bacteria. The purified phage strain was subjected to genome sequencing analysis and morphological observation. The optimal multiplicity of infection (MOI), the one-step growth curve, temperature stability, and pH stability were analyzed. Phage VA5 was observed to have a long tail. Whole-genome sequencing revealed that the genome was circular dsDNA, with 35,866 bp length and 46% G+C content. The optimal MOI was 1, the incubation period was 20 min, the outbreak period was 30 min, and the cleavage amount was 92.26 PFU/cell. The phage showed good activity at −20 °C, 70 °C, and pH 2–10. Moreover, the phage VA5 exhibited significant inhibitory effects on *V. alginolyticus*-infected shrimp culture. The isolated phage VA5 has a wide range of host bacteria and is a good candidate for biological control of pathogenic bacteria.

## 1. Introduction

*Vibrio alginolyticus* is a facultative anaerobic and halophilic gram-negative bacterium without capsule and spores. Its biological characteristics are very similar to those of *Vibrio parahaemolyticus* [1,2], and it is a conditioned pathogen that exists widely in marine water [3,4,5]. *V. alginolyticus* is not only pathogenic to fish, shrimp, and shellfish; it can also cause harm to human health by contaminating food [2,5,6]. Under the condition of the declining immune function of aquaculture animals [2], alginolytic vibriosis disease caused by *V. alginolyticus* is prone to outbreaks, especially in the summer, and often causes huge economic losses to the aquaculture industry.

A bacteriophage is a prokaryotic virus that infects bacteria. A large number and variety of phages can survive in various ecological environments [7,8]. They have the advantages of high specificity, strong isolation, replication, and fast development; and due to their strong ability to adapt to drug-resistant bacteria, they go through fewer adverse reactions during treatments [7,9,10,11]. For a long time, antibiotics have been used as a medical treatment to prevent and control bacterial diseases in aquaculture; however, due to the excessive and reckless use of antibiotics, the problem of bacterial resistance has become increasingly apparent [12,13]. Therefore, it is crucial to find better biological control methods. Oliveira et al. [14] proved through experiments on animals that phages can effectively control various types of infections, caused by different pathogens, in animals.Yang et al. [15] isolated phage LPST144 against *Salmonella typhimurium* (ATCC13311) from wastewater that completely inhibited the growth of host bacteria within 7 days. At a multiplicity of infection of 0.01-1000, the growth of the host bacteria was completely inhibited within 7 hours. At the same time, no genes related to lysozyme, antibiotic resistance and virulence factors were found, and it also had strong viability under low temperature and strong alkaline conditions, which has good potential for controlling Salmonella in the food industry. The *Escherichia coli* phage vB_EcoM_C2-3 was isolated from sewage by Maganha et al. [16]. This phage was found to be a new member of the T4-like phage, and its genomic characteristics provided valuable insights for the use of potential biocontrol agents to control multi-drug resistant *E. coli*.

Bacteriophages can serve as suitable substitutes for antibiotics in the treatment of bacterial diseases and have become a trending topic in contemporary research [17]. In this study, the phage was isolated and identified using *V. alginolyticus* as the host. Its morphology, basic biological characteristics and bacteriostatic effects were investigated. Also, the application prospects of phage VA5 were studied to control *V. alginolyticus* infection in shrimp.

## 2. Materials and Methods

### 2.1. Materials

#### 2.1.1. Samples and Strains

The samples were collected from seafood aquaculture water and aquaculture environmental effluent from the aquaculture market in Haizhou Middle Road, Lianyungang City. Collected samples were stored at 4 °C. Shrimp (*Litopenaeus vannamei*) were purchased from the aqua-market. The *V. alginolyticus* (CICC 10889) strain was procured from Jiangsu Key Laboratory of Marine Biological Resources and Environment, and used for bacteriophage proliferation and preservation. The strains used in the experiments, namely, *Vibrio parahaemolyticus*, *Vibrio cholerae*, *Vibrio vulnificus*, *Pseudomonas aeruginosa*, *Aeromonas sobria*, *Pseudomonas fluorescens*, *Aeromonas salmonicida*, *Aeromonas hydrophila*, *Lactobacillus* (*rhamnosus*), *Bacillus subtilis*, *Edwardsiella lentus*, and *Escherichia coli*, stored in our laboratory.

#### 2.1.2. Reagents and Media

Agar, tryptone, yeast powder, NaCl, DNase I, RNase A, protease K, and primer set 27F/1492R were purchased from Shanghai Sangon Bioengineering Co., Ltd. (Shanghai, China). The SM buffer contained 5.8 g NaCl, 2 g MgSO_4_·7H_2_O, 0.1 g gelatin, 50 mL 1 mol/L pH tris-HCl, and 1 L distilled water, and was sterilized. The bacteriophage buffer contained 50 mmol/L Tris, 150 mmol/L NaCl, 10 mmol/L anhydrous MgCl_2_, and 2 mmol/L CaCl_2_; the pH was set at 7.5 and the buffer was sterilized. A viral genomic DNA/RNA extraction kit and a centrifugal column bacterial genomic DNA extraction kit were purchased from Tiangen Biochemical Technology (Beijing) Co., Ltd. (Beijing, China).

### 2.2. Methods

#### 2.2.1. Phage Isolation and Screening

Thirty milliliters of each sample was taken in 50 mL centrifuge tube and centrifuged at 9803× *g* for 10 min. Supernatant was collected into a sterile dry conical bottle, and then 15 mL phage buffer and 1 mL log-growing host *Vibrio alginolyticus* solution were added (A600 = 0.55~0.6) into it. After mixing, the culture was incubated at 37 °C and 140 rpm for 48 h. After incubation, the culture medium was filtered through a 0.22 μL filter membrane, and the presence of specific phages of host bacteria in the culture medium was detected using the double-layer agar plate method [18]. Two hundred microliters of logarithmic host bacteria solution was taken in a 10 mL centrifuge tube, followed by the addition of 200 μL filtrate on an aseptic super-clean table. After 15 min, 5 mL semi-solid medium incubated at 55 °C was added in the tube. The tube was closed with a lid and upturned 3 times. After upturning and mixing, the content of the tube was poured onto a solid medium, and the medium was allowed to solidify completely. Subsequently, the plates were placed upside down in a biochemical incubator and incubated for 48 h at 37 °C. After incubation, the presence of phage on the plates was confirmed by observing the plaques or clear zones.

#### 2.2.2. Bacteriophage Purification and Titer Determination

The bacteriophage was purified using the double-plate method [18]. A transparent zone was selected to pick pure phages and place them in fresh logarithmic host bacteria solution (OD_600nm_ = 0.55~0.6). The inoculated bacterial solution was left at room temperature for 20 min and then incubated for 20 min at 180 rpm and 37 °C. The 100 μL culture solution was placed in a 2 mL centrifuge tube and serially diluted ten times using SM buffer. Subsequently, a 200 μL bacterial solution and 200 μL diluent were taken in a 10 mL centrifuge tube and mixed, and the tube was left on a sterile super-clean table for 15 min. Afterwards, 5 mL semi-solid medium heated at 55 °C was immediately poured onto the prepared Luria-Bertani (LB) solid agar plate to make a double plate. The double-layer plates were then incubated at a constant temperature of 37 °C for 48 h. After incubation, the growth of transparent circles was observed. The above operations were repeated 5–6 times until distinct and evenly distributed phage spots of the same size appeared. After obtaining purified phages, the titer of the phage proliferation solution was determined using the double-layer plate method.

The phage solution was diluted ten times with SM buffer. Then, 200 μL of *V. algaelyticus* (OD_600nm_ = 0.55~0.6) and 200 μL of bacteriophage fluid were added into a centrifuge tube (10 mL) and mixed well. The mixture was added to 5 mL of LB semi-solid medium containing 0.65% agar, and immediately poured into the prepared LB solid agar plate to make a double-layer plate, which was allowed to solidify at room temperature. The double-layer plates were then incubated at 37 °C. Three parallel samples were taken for each dilution, and plaques were counted the next day. The final number of plaques was the average of the three parallel experiments. The phage titer was calculated as follows [18]:Phage titer (PFU/mL) = number of plaques × dilution × 10(1)

#### 2.2.3. Identification of Phage Morphology

Ten μL (>10^7^ pfu/mL) of phage proliferation solution was dropped onto disposable PE gloves, and the front of the copper mesh was kept in contact with the phage proliferation solution for 10–15 min. Negative staining with 2% (volume fraction) uranium acetate (PTA) was conducted for 10 min and repeated five times. After drying the copper mesh at room temperature, the morphology of the bacteriophage was observed and photographed using a transmission electron microscope.

#### 2.2.4. Genome-Wide and Phylogenetic Analysis

The viral genome was extracted using the Viral Genome DNA/RNA Extraction Kit. The phage genome samples were sent to Sangon Biotech for whole-genome sequencing via the Illumina sequencing platform. The quality of the raw data was assessed using FastQC, quality shearing using Trimmomatic [19], splicing of second-generation sequencing data using SPAdes [20], prediction of gene elements using Prokka, and identification of gene proteins using the RepeatMasker to identify repetitive sequences on the genome [21]. Promoter prediction was performed using softberry software (http://www.softberry.com/ (accessed on 23 July 2023)) [22]. ORF prediction was performed on the raw data using the NCBI server (https://www.ncbi.nlm.nih.gov/cdd/ (accessed on 23 July 2023)). Phage genomes were downloaded (https://www.ncbi.nlm.nih.gov/ (accessed on 23 July 2023)) and aligned using MEGA 11software. A phylogenetic tree was generated using the neighbor-joining method with 1000 bootstrap replications.

#### 2.2.5. Determination of Optimal Multiple Infection of Phage

To determine the optimal multiplicity of infection (MOI) of the phage, the method of Qu et al. [23] was used with slight modifications. The phage and host bacteria were mixed according to different MOI ratios, ranging from 0.01 to 100, and cultured in LB liquid medium for 6 h. The titer was then measured. The host bacteria count method was used to inoculate the host bacteria at an inoculum concentration of 1%. The culture was incubated at 37 °C and 180 rpm for 8 h. Sample collection was started after 2 h of incubation. One milliliter of bacterial solution was collected every 1 h and serially diluted 10-fold to measure the OD600 and plot a standard curve.

According to the MOI ratio of 100, 10, 1, 0.1, 0.01, 200 μL of culture solution and 200 μL of log-phase host bacterial solution were placed in a 10 mL centrifuge tube, mixed well, and left on a sterile ultra-clean table for 15 min. Afterwards, the mixed culture was added to 5 mL of semi-solid medium heated at 55 °C. The medium containing the culture mixture was mixed evenly and poured onto LB solid medium. Fully solidified plates were placed upside down in a biochemical incubator, and incubated at 37 °C for 48 h. After incubation, the growth of the transparent circle was observed. Each experiment was replicated three times.

#### 2.2.6. One-Step Growth Curve of Phage

According to the optimal number of infections, the phage and host bacteria were mixed and cultured at 37 °C and 200 rpm for 20 min, and centrifuged at 4 °C at 15,777× *g* for 10 min. The supernatant was discarded, and the precipitate was suspended in 2 mL LB broth medium, preheated to 37 °C. The culture was incubated at 37 °C and 200 rpm. During incubation, samples were collected every 10 min and the titer was measured using the double-layer agar plate method. Three parallel experiments were carried out in each group to determine the incubation period, burst period and cleavage amount of phage. The culture time was 180 min.

#### 2.2.7. Effects of Temperature, Ultraviolet Irradiation, and pH on Phage Activity

In order to study the stability of phage VA5 under ultraviolet (UV) irradiation, different temperatures, and pH values, the method described by Yang et al. [22] was used with slight modifications. The phage was exposed to ultraviolet rays, and samples were collected every 10 min for 60 min. Titer was measured and the influence of ultraviolet rays was determined. To examine the effect of temperature, the phage suspension (3.32 × 10^6^ PFU/mL) was incubated in a water bath at −20 °C, 4 °C, 25 °C, 37 °C, 50 °C, and 70 °C for 0 min, 40 min, and 120 min, and the titers were measured. Nine milliliters of LB liquid medium with different pH values (1, 2, 3, 4, 5, 6, 7, 8, 9, 10, 11, 12) were added into sterilized test tubes and cultured at 37 °C and 180 rpm for 6 h. The titers were measured, and the influence of pH was evaluated. All UV, temperature, and pH experiments were conducted using three parallel samples.

#### 2.2.8. Bacteriophage Inhibitory Activity

The experimental method of Li et al. [24] was used, with slight modification, to analyze the inhibitory activity of the phage. Shrimp of uniform size, purchased from the aquatic product market, were temporarily reared. Shrimp with good vitality were selected and divided into six groups, with 15 shrimp being continuously oxygenated in each group. The directly cultured shrimps (without any treatment) were used as the control group (CK). In the phage group (BP), phage suspension was sprayed onto the shrimp injected with *V. alginlyticus.* In the phage control group (P), phage liquid was sprayed onto the shrimp. In the *V. alginlyticus* group (VA), no spraying was applied to the shrimp injected with *V. alginlyticus*. Each tank of 15 shrimp was continuously oxygenated.

VA group: The VA injection point was the muscle of the second body segment of shrimps. The needle was directly injected into the subcutaneous tissue of the peritoneum for microinjection. *V. alginolyticus* in logarithmic phase was used at a concentration of 1.68 × 10^8^ (cfu/mL), and the phage liquid potency was 1 × 10^7^ (pfu/mL). Ten μL of bacterial solution was fully injected into the shrimp and allowed to diffuse into the body for 1–2 s. Then, the shrimp were put into the incubator. Group BP: The shrimp injected with *V. alginolyticus* solution were divided into three groups, and after 1 h, 0.8 mL, 1.5 mL, and 2 mL of phage solution were sprayed onto the shrimp in the three groups, respectively. Group P: After spraying the phage solution, the physiological responses of the shrimp were observed at 0 h, 3 h, 6 h, 12 h, 24 h, and 36 h, and the mortality rate was calculated [24].
Mortality rate (%) = dead shrimp tail number/initial tail number × 100%(2)

#### 2.2.9. Determination of Host Bacteria Range

The activated phage VA5 was serially diluted 10 times with SM buffer solution. Two hundred microliters of each dilution gradient was placed in a 10 mL centrifuge tube, and 200 μL of bacterial solution was added, as shown in Table 1. After keeping the centrifuge tube for 15 min on a super-clean table, a semi-solid medium containing 0.65% LB was added to the tube. The mixture was immediately poured onto the prepared LB solid agar plate to make a double plate. After solidification at room temperature, double-layer plates were incubated in an incubator at 37 °C for 48 h. The clarity of the formed plaque was observed to confirm the phage infection in the tested strain. A clear plaque indicates that the bacteria was the host, and a fuzzy plaque or no plaque indicates that the bacteria cannot be lysed.

#### 2.2.10. Statistics and Analysis of Data

Origin2018 and Graphpad prism 8.0.2 software were used for data analysis and processing, and the *t*-test was used for analyzing the significance of the differences (*p <* 0.05). Each experiment was conducted with three parallel samples.

## 3. Results

### 3.1. Isolation and Purification of Phages and Morphological Characteristics of Plaque

*V. alginolyticus* was used as the host bacteria to isolate phages using the double-layer agar plate method. Phages were cultured in a biochemical incubator at 37 °C for 48 h. Afterwards, large and transparent phage plaques on the plate were selected for sub-culturing. The phages were purified and sub-cultured repeatedly until all the plaques on the plate were roughly the same shape and size. Thus, a pure phage strain was isolated and named phage VA5. The morphology of the plaques is shown in Figure 1. Phage plaques showed high transparency and a regular round shape. A transmission electron microscope revealed that phage VA5 consisted of a head and a tail and belonged to the phage of Siphoviridae.

### 3.2. Whole-Genome Analysis of Phage

The phage VA5 genome sequence was deposited in the Banklt database under accession number OR754009. The phage genome was 35,866 bp in length, making a circular dsDNA with 46% G+C content (Figure 2). A total 524 ORFs of phage genome were predicted, with an average length of 1265 bp. There were 91 protein coding genes, including 10 genes related to unknown functional proteins. Ninety-one potential promoters were queried using the BPROM program (Softberry). In addition, the presence of tRNA genes in VA5 implied that VA5 replication was not entirely dependent on the host translation machinery [25].

### 3.3. Phylogenetic Analysis

A phylogenetic tree is a diagram that illustrates the origin and evolution of species. In Figure 3, it can be observed that phage VA5 and Vibrio phage ICP2 (accession n. NC 015158.1) are grouped together on a single branch, suggesting that phage VA5 is one of the Vibrio phages. The homology was 45.72% between VA5 and ICP2 [26]. Therefore, phage VA5 might be a novel strain of Vibrio phage.

### 3.4. The Optimal MOI and One-Step Growth Curve Determination

The results are shown in Figure 4A. The optimal MOI of the phage was found to be 1.

The one-step growth curve is an important index with which to measure the lytic ability of a phage. It can be seen from Figure 4B that the incubation period of the phage was 20 min and the outbreak period was 30 min. The amount of phage lysis was calculated by dividing the titer of the phage at the end of lysis with the concentration of the host bacteria at the initial stage of infection [3]. The amount of phage VA5 lysis was calculated to be 92.26 PFU/cell.

### 3.5. Determination of Phage Stability under UV Irradiation, Temperature, and pH

Under UV irradiation, the titer of phage VA5 decreased by 1 titer in the first 10 min, and then continued to decrease every 10 min (Figure 5A). After 60 min of UV exposure, the titer decreased significantly. In the thermal stability test, phage VA5 maintained good activity in each temperature range for a long time. However, at 50 °C and 70 °C, the titer of phage VA5 decreased by 2 and 3 titers, respectively, with the increase in culture time (Figure 5C). The results of the pH tolerance test showed that phage VA5 had good activity in the range of pH 2–10 and had tolerance for a wide range of pH. However, the activity of phage VA5 decreased significantly under acidic conditions below pH 2 and alkaline environments above pH 11 (Figure 5B).

### 3.6. Determination of the Range of Host Bacteria

Phage VA5’s host bacteria range is shown in Table 1. Phage VA5 showed lytic activity in 11 strains of bacteria. The lytic activity of phage VA5 was especially stronger in *Vibrio parahaemolyticus* and *Pseudomonas fluorescens*, while no effects of phage VA5 were observed in *Pseudomonas aeruginosa* and *Bacillus carbonmaggots.*

### 3.7. Antibacterial Properties of Phages

The cumulative mortality of shrimp after *V. alginolyticus* infection is shown in Table 1 and Figure 6. At 12 h, the mortality rate in the BP group significantly reduced. At 24 h, the rate of death in the BP group was significantly lower than that in the VA group. The lowest mortality rate in the BP group was observed for the shrimp sprayed with 2 mL of bacteriophage solution. In group P, which was not infected with *V. alginolyticus*, the mortality of shrimp was lower than that in group CK. At the same time, shrimp infected with *V. alginolyticus* were observed to have yellow gills, pleopods, red pereiopods, and broken antennas (Figure 7).

## 4. Discussion

Samples in this experiment were collected from sewage from the aquatic product wholesale market and from the breeding water of the aquatic product. It was difficult to isolate phages from the aquaculture water samples, which may be due to the use of drugs and the strict supervision during the aquaculture process.

In this study, a phage strain of *V. alginolyticus* was successfully isolated from the water of aquaculture and named VA5. The large and transparent plaques on double-layer agar plates indicated that phage VA5 was a potent phage. There was one coding tRNA sequence in the phage genome. Previously, it has been shown that the tRNA in the phage genome can make up for the lack of tRNA in the host bacteria, and that the phage can improve the efficiency with which proteins are translated, compared to alternatives such as cleavage enzymes [27]. The phage containing the tRNA gene was reported to have a wide host range rather than a specific host [28].

The optimal MOI value of phage VA5 was 1, which indicated that the phage could achieve a good bactericidal effect at lower concentration. However, in this study, the optimal MOI value of a phage required further adjustment. Increasing the optimal MOI of a phage can more effectively reduce the cost of phage application and is conducive to the large-scale production and application of phage products [24].

The one-step growth curve results showed that the incubation period of the phage was 20 min, while its outbreak period was 30 min. The burst volume of 92.26 PFU/cell indicates that phage VA5 can produce a large number of progeny phages in a short period of time and that it can more effectively act on the host bacteria [5]. The survival rate of phage VA5 was found to be optimal in the pH range of 2–10 and temperature range of −20 °C to 70 °C. This indicates that phage VA5 has a strong tolerance to acidic and alkaline environments and is thermally stable. The good activity shown by the phage under a wide range of conditions proves that bacteriophage phage VA5 can survive in a harsh environment and has great application potential.

Antibacterial experiments showed that phage VA5 inhibited the activity of *V. alginolyticus* well and reduced the mortality of shrimp. At 24 h, the mortality rate in BP group was significantly lower than that of the VA group, with the lowest mortality rate shown by BP shrimp sprayed with 2 mL of bacteriophage solution. This indicates that, in the presence of a certain phage titer, an appropriate increase in the dose of phage suspension could reduce the mortality rate of shrimp [5,24]. Group P, which was not infected with *V. alginolyticus*, showed a lower mortality rate than group CK. These findings prove that phage VA5 has a good antibacterial effect against *V. alginolyticus* and that it can also inhibit the growth of other micro-organisms and reduce the mortality of shrimp.

Furthermore, it was found that the phage has a lytic effect on various pathogenic strains, as shown in Table 2. The tRNA gene in phage VA5 may play a pivotal role and it can improve the effect of tRNA on the host bacteria’s ability to act [25,29]. Alternatively, phage VA5 with a tRNA gene can be used as a biosensor to detect pathogenic strains through a rapid detection method based on bacterial cell lysis [22,30].

Further in-depth research is required on the similarities between the tRNA gene and cleavable pathogenic bacteria. Moreover, the cleavage effect of phage VA5 on pathogenic bacteria needs to be improved by changing its structure to reduce the impact of pathogenic bacteria on aquaculture and improve the application potential of this bacteriophage.

## 5. Conclusions

When phage VA5 was isolated using *V. alginolyticus*, it was found to consist of a head and a tail, to belong to the family Longtaviridae, and to be a Vibrio phage. The full length of the phage VA5 genome was 35,866 bp, and it was circular dsDNA with 46% G+C content. Its MOI was 1. It had a latency period of 20 min and a lysis volume of 92.26 PFU/cell. It showed good tolerance for temperature and pH variations. Furthermore, phage VA5 could significantly reduce the mortality rate of shrimp *(L. vannamei*) infected with *V. alginolyticus*. Meanwhile, it was able to exhibit lysis activity in 11 pathogenic bacteria, which showed the potential of its application to control multiple pathogens.

## Figures and Tables

**Figure 1 microorganisms-11-02822-f001:**
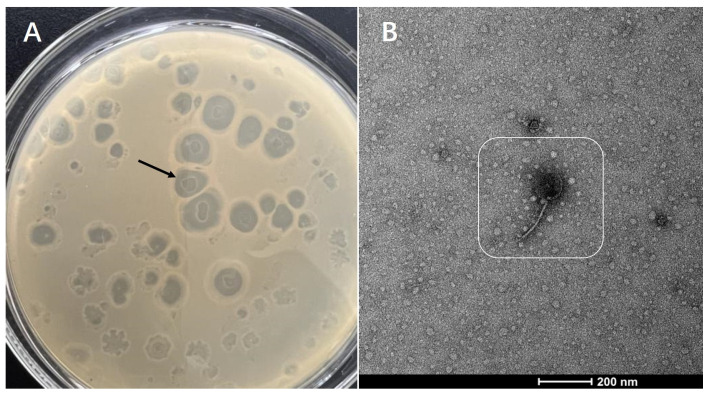
Identification of phages: (**A**) Phage plaques growing on the layer of *V. alginolyticus*; (**B**) Phage morphology under transmission electron microscope.

**Figure 2 microorganisms-11-02822-f002:**
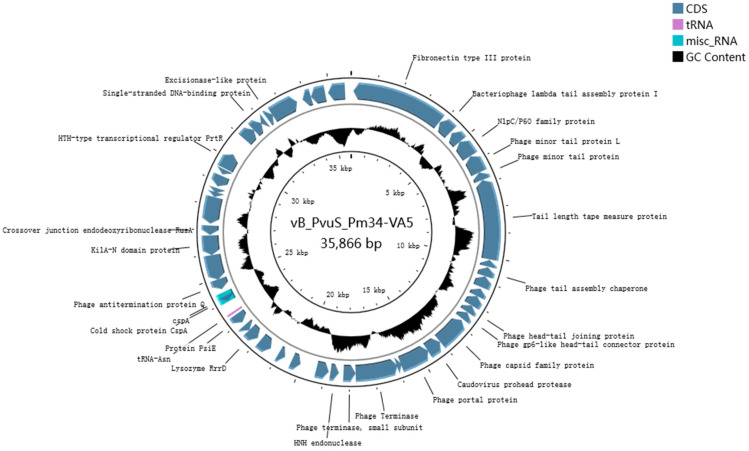
Phage VA5 genome map. The predicted coding region has been represented by an arrow indicating the direction of transcription. The black circle indicates the G+C content.

**Figure 3 microorganisms-11-02822-f003:**
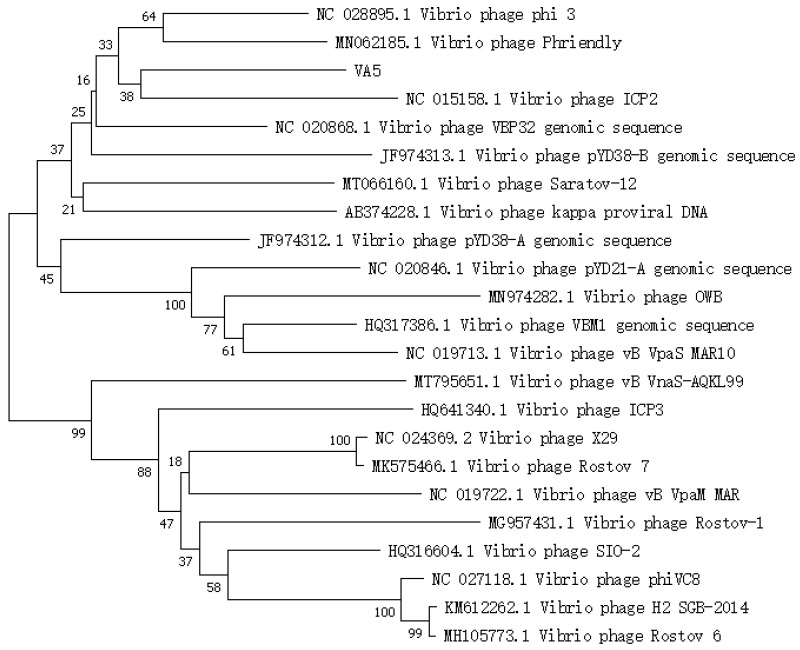
Phylogenetic tree of phage VA5.

**Figure 4 microorganisms-11-02822-f004:**
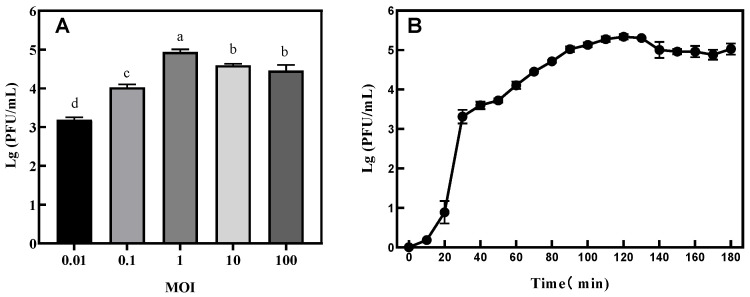
(**A**) Determination of optimal multiplicity of infection (MOI) and (**B**) one-step growth curve of phage VA5. Different letters indicate significant differences between MOIs. (*p* < 0.05).

**Figure 5 microorganisms-11-02822-f005:**
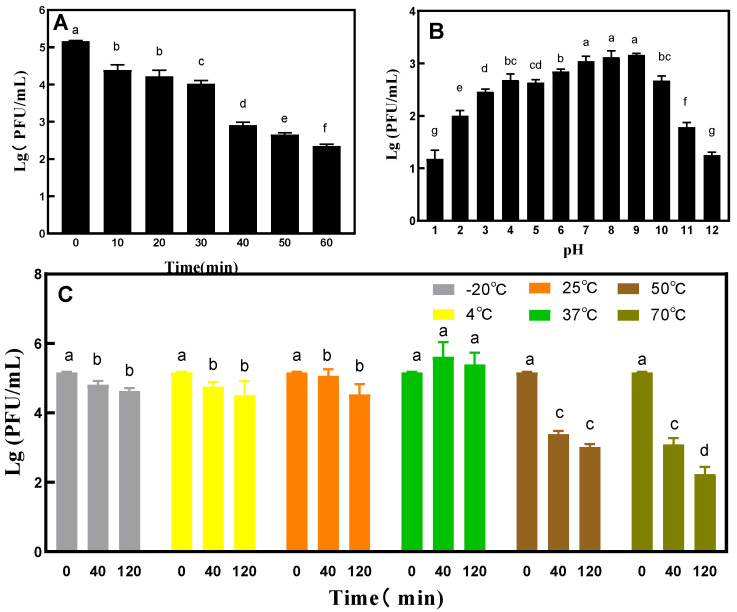
General characterizations of phage VA5: (**A**) UV tolerance, (**B**) pH tolerance and (**C**) Temperature tolerance. Different letters indicate significant differences, and double letters indicate significant differences between the two groups. (*p* < 0.05).

**Figure 6 microorganisms-11-02822-f006:**
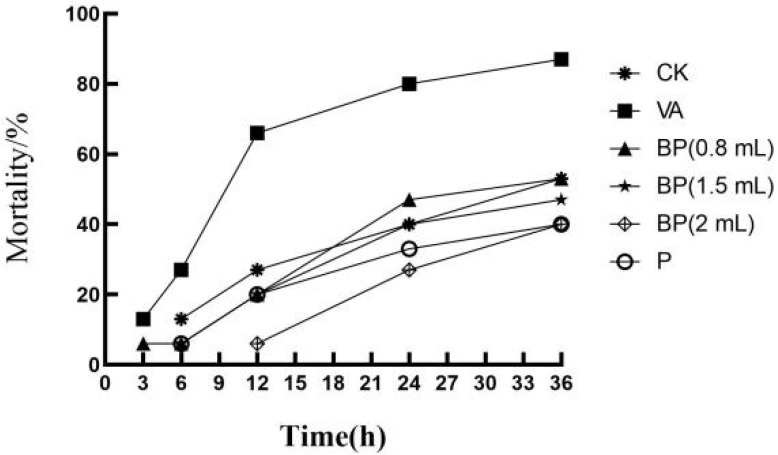
Cumulative mortality of shrimp after *V. alginolyticus* infection.

**Figure 7 microorganisms-11-02822-f007:**
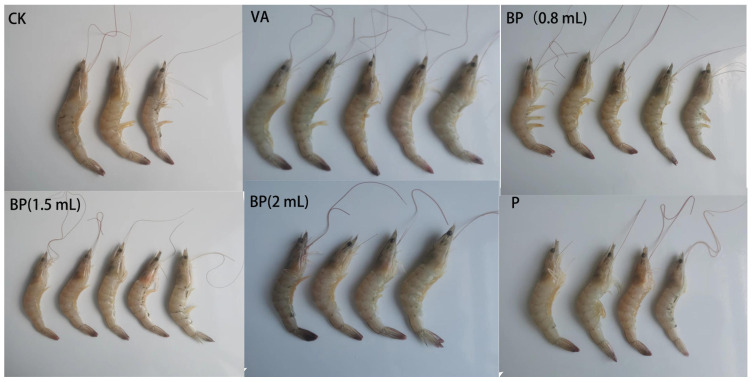
Cumulative mortality of shrimp infected with *V. alginolyticus*. CK: control group, VA: In *V. alginlyticus* group, BP (0.8 mL): VA + phage group, BP (1.5 mL): VA + phage group, BP (2 mL): VA + phage group, P: phage group.

**Table 1 microorganisms-11-02822-t001:** Range of host bacteria for phage VA5.

Bacteria	Plaque Formation by
	Phage VA5
*Vibrio parahaemolyticus*	*****
*Vibrio cholerae*	***
*Vibrio vulnificus*	**
*Pseudomonas aeruginosa*	\
*Aeromonas sobria*	**
*Pseudomonas fluorescens*	*****
*Aeromonas salmonicida*	***
*Aeromonas hydrophila*	*
*Lactobacillus (rhamnosus)*	*
*Bacillus subtilis*	**
*Edwardsiella lentus*	**
*Escherichia coli*	*

Note: * represents the amount of phage plaques formed; the more *, the more phage plaques, indicating a higher capacity for phage lysis; ***** represents the maximum value of phage plaques, while \ represents no plaque formation.

**Table 2 microorganisms-11-02822-t002:** Cumulative mortality of shrimp after *V. alginolyticus* infection.

Time (h)	Mortality Rate (%)					
	CK	VA	BP (0.8 mL)	BP (1.5 mL)	BP (2 mL)	P
0	0	0	0	0	0	0
3	0	13	6	0	0	0
6	13	27	6	6	0	6
12	27	66	20	20	6	20
24	40	80	47	40	27	33
36	53	87	53	47	40	40

Note: CK: control group, VA: in V. alginlyticus group, BP (0.8 mL): in phage group, BP (1.5 mL): in phage group, BP (2 mL): in phage group, P: in phage control group.

## Data Availability

The datasets generated during and/or analyzed during the current study are available from the corresponding author on reasonable request.

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
