# Peer review of "Isolation and Characterization of Bacteriophage VA5 against Vibrio alginolyticus"

_microorganisms, 2023, doi:10.3390/microorganisms11122822_

Round 1
Reviewer 1 Report
Comments and Suggestions for Authors
The authors conducted a whole-genome sequence and characterization of the phage infecting Vibrio alginolyticus, which they had newly isolated. The inhibition of lethal activity by this phage in V. alginolyticus infection experiments on shrimp was also examined. The use of phage to suppress the virulence of V. alginolyticus is important as a very effective antibiotic-free antimicrobial method in shrimp farming.
In p.3, L115, you stated that phage titers were >10^7/mL, but in Figures 3 and 4, the highest phage titers produced were 10^5/mL. Describe the concentration method of this phage to 10^7/mL.
In p. 4, L167, why is V. alginolyticus injected into the body with a needle, but phage is sprayed outside the body? Has it been confirmed that the phage was infected with V. alginolyticus? Also, the injection volume of 10uL of V. alginolyticus or the spray volume of phage VA5 was described, but please describe their concentrations, respectively.
In P. 4, L174, after injection of V. alginolyticus, the phage was spray infected, how long was the infection time? Also, after infecting the phage, was the sprayed phage solution rinsed off when returning the shrimps to the aquarium?
In P. 5, L210, you analyzed the whole genome sequence of bacteriophage VA5. Have you submitted this data to public databases such as GenBank? If submitted, please provide its accession NO. If there were 91 protein-coding genes and 91 promoter sequences in the genome, does that mean that they can be expressed independently of each other?
In Table 1, discuss why the lethality rate after 6 hours of spraying is lower in the phage sprayed group (P) than in the control group (CK). Also, please describe the volume of phage sprayed at this time.
In Table 2, the susceptibility of phage VA5 was examined in various bacterial species other than V. alginolyticus, but there were no results of similar tests with different V. alginolyticus isolates. Were any V. alginolyticus isolates similarly phage-sensitive? With the sensitivity of phage VA5 to various species of bacteria, have you investigated the correlation with the DNA methylation activity that each species possesses? One possible method of verification is, for example, to determine whether E. coli is as susceptible to phage VA5 as it is when reinfected with phage prepared from infected E. coli, which has a low susceptibility the authors mentioned.
P. 1, L30, Vibrio parahaemolyticus -> In italics (and many other parts of the text)
P. 1, L39, high specificity, specificity -> Leave one or the other behind.
P. 1, L39, self-replication -> Phage multiply themselves?
P. 8, L271, Bacillus carbonmagotts -> I could not find such a species in the database. Nor was it listed in Table 2.
Table 2., Anthrax bacillus -> Bacillus anthracis
Comments on the Quality of English Language
I don't think there are any major problems.
Reviewer 2 Report
Comments and Suggestions for Authors
In this study the inhibitory effects of a bacteriophage isolated from aquaculture-associated resource were examined. This manuscript has several major flaws, and the following corrections should be made in order to achieve the standard of publication:
Line 1. The title of the manuscript is suggested to be changed to “Isolation and Characterization of Bacteriohage VA5 against Vibrio alginolyticus”.
Line 21. Please revise “lysed amount”.
Line 31. “Vibrio alginolyticus” should be italic.
Lines 32-33. Please revise “, but it can also cause”.
Line 34. Please rewrite “alginolytic vibriosis disease”.
Line 48. Please revise “pulsentery”.
Lines 49, 52. “Escherichia coli” should be italic.
Line 49. Please change “isolated” to “was isolated”.
Line 54. Please change “hot topic”.
Lines 62-63. Please provide more information about “the aqua-market”.
Line 63. Please specify the source of other bacterial strains used in this study.
Line 77. Thirty not 30. Please correct this throughout the manuscript (lines 83, 130, 156, 181, 212).
Lines 77, 142. “×g” should be italic.
Lines 105-107. Please revise “200 μL of V. alginolyticus solution (OD600nm=0.55~0.6) was mixed with 200 μL of V. alginolyticus solution (OD600nm=0.55~0.6) and 5 mL of LB semi-solid medium”.
Line 122-125. Please provide the accession number of the phage genome if it has already been deposited in GenBank or other related databases.
Line 132. Please change “draw” to “plot”.
Line 133-134. Please rewrite “200 μL of culture solution and 200 μL of host bacterial solution”.
Line 143. Please replace “liquid” with “broth”.
Line 154. Should be “3.32×106”.
Lines 162-178. These two paragraphs should be rewritten as there are confusing sentences (experimental groups have not been well described, number of shrimps used per treatment and tank is missing, there is no information whether shrimps were anesthetized before injection).
Line 180. This paragraph and the obtained results are confusing as none of bacteria used have been isolated from the host.
Line 183. “Table 2” is correct.
Lines 188-189. Please revise “was the host”.
Line 192. I have some concerns about the statistical tests (please specify if one-way ANOVA was used). Also, the significant differences have not been shown in the figures correctly.
Line 203. Please change “the long” to “a long”.
Lines 226-228. This sentence should be moved to the section M & M. No reference should be provided here.
Lines 252-253. Please rewrite “swimming feet” and “brittle and easily breakable”.
Line 254. Please specify the reason why there were many mortalities in the control group (CK).
Line 261. The legend of Figure 6 should be rewritten. There is no information about the first set of images included in this figure.
Lines 263, 271. Please specify which one of these isolates were used and provide their correct scientific names “Anthrax bacillus” or “Bacillus carbonmaggots”.
Line 270. “Vibrio parahaemolyticus and Pseudomonas fluorescens” should be italic.
Line 273. Please revise “sewer sewage”.
Line 306. In this study, V. alginolyticus was injected into the muscle of shrimps, and there was no direct exposure of the bacteriophage used with the pathogen. What could be the possible protective effects? Please specify in this paragraph.
Line 309. Please change “decisive role” to “pivotal role”.
Comments on the Quality of English Language
Please see the comments.
Reviewer 3 Report
Comments and Suggestions for Authors
1- Pls change Vibrio alginolyticus; Bacteriophage in the keywords as it already presents in the title.
2- Line 30: Vibrio parahaemolyticus should be italic.
3- Line 34: there is no disease called alginolytic vibriosis but vibriosis caused by…..
4- Line 39: remove the repeated word specificity.
5- English editing is required.
6- Line 49: Escherichia coli should be italicized.
7- Line 52: E. coli should be italicized.
8- Shrimp (Litopenaeus vannamei) was purchased from aqua-market, from a market or farm to be live and viable, the number, size, and weight should be mentioned.
9- Line 79: the type of host bacteria should be mentioned.
10- Line 88: replace were put upside down with incubated in an inverted position.
11- Line 106 to line 109: 200 μL of V. alginolyticus solution (OD600nm=0.55~0.6) was mixed with 200 μL of V. alginolyticus solution (OD600nm=0.55~0.6) and 5 mL of LB semi-solid medium, what that mean? Please correct.
12- Is it true or effective to spray the bacteriophage on shrimp can it penetrate the cuticle? Is the phage effective in vivo or more effective if was used in water?
13- Figure 2: The data on the figure should be large enough to be read.
14- What are the causes of shrimp mortality in the control (non-infected non-treated group CK) and for phage-treated non-infected group P? Are the results logical?
15- Range of host bacteria for phage: mention used bacterial isolates and their names, reference numbers, and sources in the materials and methods section.
16- Furthermore, it was found that the phage had a lytic effect on various pathogenic strains, as shown in Table 1: Table 2 not 1.
17- Some parts of the conclusion should be transferred to the results from lines: 319 – 327.
18- The conclusion is lengthy please deduce in only 5 lines maximum.
Comments on the Quality of English LanguageEnglish language should be edited, many paragraphs should be deduced
Reviewer 4 Report
Comments and Suggestions for Authors
This paper presents the findings from an experiment to Isolate and Identificate of Phage VA5 against Vibrio alginolyticus and its characterization. The authors purified the phage strain and subjected to genome sequencing analysis and morphological observation, as well as was determined the Optimal multiplicity of infection (MOI), one-step growth curve, temperature stability, and pH stability were analyzed. Phage VA5 was observed to have a long-tail and was study against many different bacteria. I believe there is much merit to the data presented in this paper and the findings are very consistent, and in the future this method can represent important tool for sanitary management of aquaculture. However, minor problems were observed in the manuscript, fact that compromises its publication thereby. I would encourage the authors to adjust the manuscript to publish it in this important journal. In this context, the following comments should be addressed:
- Line 49 and 52 (introduction): Escherichia coli should in italic.
- Line 154 in phage suspension should be ‘6” must be superscript.
- Lines 162 to 168 should be better explained (it is not clear the shrimp’s groups, rephrased).
- Table 1 (line 254) captions should explain “CK, VA, BP and P, groups” (the table has to be self-explanatory). The same should be done in Figure 5 and 6.
- Discussion section should be formatted (text need to be justified)
- Line 309 is Table 1 or 2?
-Line 334 Vibrio parahaemolyticus and Pseudomonas fluorescens should be in italic.
Round 2
Reviewer 2 Report
Comments and Suggestions for Authors
The authors have provided nearly all the revisions requested, and the manuscript can be published in the journal once the accession number of the phage genome is added to the MS.
Please also correct "Vibrio alginolyticus" in line 112.
Lines 142-143. Please revise "take" and "mix".
Reviewer 3 Report
Comments and Suggestions for Authors
thanks to all of the authors for response to my comments
Author Response
I am very grateful to the reviewers for their valuable comments and suggestions on my thesis, which made it a great improvement.